# Navigating a Pandemic: Leadership Dynamics and Challenges within Infection Prevention and Control Units in Israel

**DOI:** 10.3390/healthcare11222966

**Published:** 2023-11-15

**Authors:** Dafna Chen, Stefan Cojocaru

**Affiliations:** Department of Sociology and Social Work, Alexandru Ioan Cuza University from Iasi, 700506 Iasi, Romania; contact@stefancojocaru.ro

**Keywords:** infection prevention and control (IPC), healthcare leadership, coronavirus disease 2019 (COVID-19) pandemic, managerial challenges, self-efficacy, implementation of the IPC program

## Abstract

This study investigates the impact of the coronavirus disease 2019 (COVID-19) pandemic on leadership within infection prevention and control (IPC) units across public hospitals in Israel. Through qualitative interviews with ten IPC managers from nine hospitals, equivalent to 30% of the country’s acute care facilities, the research uncovers significant changes in managerial approaches due to the health crisis. The results reveal four main themes: (1) Enhanced managerial autonomy and leadership skills, with a noted rise in self-efficacy against the pandemic’s backdrop; (2) Shifted perceptions of IPC units by upper management, recognizing their strategic value while identifying the need for a more profound understanding of IPC operations; (3) The increased emphasis on adaptability and rapid decision-making for effective crisis management; (4) The dual effect on job satisfaction and well-being, where greater commitment coincides with risks of burnout. The study underscores the essential nature of effective IPC leadership during emergencies, highlighting the need for clear communication, prompt action, and empathetic leadership. The conclusions point to the necessity for continuous research into IPC leadership, promoting strategic advancements in management to bolster IPC units against future health threats.

## 1. Introduction

Emerging from the insights of doctoral-level research, this article delves into the profound challenges the coronavirus disease 2019 (COVID-19) pandemic posed on healthcare organizations and their leaders, especially those at the helm of infection prevention and control (IPC) units. In these tumultuous times, leadership fortified by trust and credibility has become paramount, not only in safeguarding patients and staff but also in swiftly navigating the labyrinth of ceaseless changes [1,2]. Amidst this backdrop, IPC teams found themselves on a steep learning curve, cultivating new competencies, galvanizing their units towards the adoption of novel practices, and making determinations amidst pervasive uncertainties. Trust has prominently risen as an essential component in crisis management, bolstering team morale and steadfast commitment [2,3]. This research casts a spotlight on the influence of the COVID-19 pandemic on the managerial roles in public hospitals in Israel and the vital skills that hospital IPC unit managers ought to cultivate, especially when confronted with decision-making in unpredictable and uncertain scenarios.

Fundamentally, IPC teams in hospitals aim to mitigate the risks associated with healthcare-associated infections (HAIs) and antibiotic resistance. HAIs have emerged as a grave public health concern, resulting in substantial morbidity, mortality, and worldwide economic repercussions [4,5,6,7,8,9]. While the Centers for Disease Control and Prevention (CDC) have issued guidelines since 1981, adherence remains suboptimal [5,10]. The degree of trust vested in these guidelines, coupled with the credibility of the entities promulgating them, holds substantial sway over compliance rates [2,11]. The World Health Organization (WHO) has underscored the development of globally harmonized recommendations for efficacious IPC programs to delineate evidence-driven interventions and bolster compliance. Nonetheless, IPC teams grapple with myriad challenges such as burgeoning workloads, knowledge deficits, resource constraints, and leadership gaps [4,9,12,13,14,15]. The integrity of information sources and the perceived authenticity of leadership are instrumental in managing these hurdles and can considerably shape the efficacy of IPC strategies [16,17,18].

The COVID-19 pandemic has added layers of complexity to the already challenging domain of HAIs, leading to increased rates of infections such as CLABSIs, CAUTIs, VAEs, and MRSA bacteremia, while also exacerbating the issue of antimicrobial resistance. The response of healthcare systems to these heightened challenges, particularly during pandemic conditions, reflects a pressing need for adaptability and reinforcement of HAI prevention strategies [19,20,21,22,23,24,25,26].

The purpose of this study is to examine the effects of the COVID-19 pandemic on managerial self-efficacy, autonomy, and leadership capabilities among IPC unit managers in Israeli public hospitals.

### 1.1. Literature Review

#### 1.1.1. The COVID-19 Pandemic and Acquired Infections

The COVID-19 pandemic has posed significant challenges to hospitals and healthcare facilities around the world. These challenges include increased workload and stress, changes in infection control protocols, limited resources, increased patient numbers, staffing challenges, and other operational changes that limit the implementation and effectiveness of standard IPC practices [19,20,21,22,23]. Trust in guidelines, protocols, and leadership plays a critical role in ensuring effective implementation of IPC practices and maintaining healthcare professional motivation in these challenging circumstances [1,2].

In addition, patients infected with COVID-19 may be vulnerable to other infections due to multiple comorbidities, prolonged hospitalization, and impaired immune system function. Bacterial co-pathogens are commonly identified in viral respiratory infections and are important causes of morbidity and mortality. Patients with COVID-19 have acquired secondary bacterial infections or superinfections, mainly bacteria and urinary tract infections [24,25,26].

Moreover, hospitals have seen substantial increases in central line-associated bloodstream infections (CLABSIs), catheter-associated urinary tract infections (CAUTIs), ventilator-associated event (VAEs), and MRSA bacteremia. The overuse of antimicrobial agents has contributed to the increased prevalence of antimicrobial resistance [19,21,22].

#### 1.1.2. The National IPC Programs (NPIPC) and the Presentation of the Gaps

The national IPC programs (NPIPC) developed by WHO and a team of experts prioritize the development of global recommendations based on the eight core components of effective IPC programs, which can be prioritized depending on the context, previous achievements, and identified gaps. The core components include IPC programs, IPC guidelines, IPC education and training, follow-ups, multimodal strategies, monitoring/review of IPC procedures and feedback, workload, personnel and bed occupancy (acute health facility only), and built environment, materials, and equipment for IPC at the hospital level [21].

However, gaps were found in the implementation of IPC programs, which include lack of guidelines in general and in particular for complex interventions in complex organizations; lack of effectiveness, independence, and applicability; weak behavioral components aimed at removing and addressing environmental, organizational, and individual barriers; lack of a systematic approach that ensures the continuity of the implementation of guidelines over time; limited resources; lack of awareness and understanding; lack of infrastructure and equipment; complex healthcare systems; lack of standardization; poor compliance; human error; patient factors; environmental factors; poor communication and coordination; lack of leadership and support; and lack of research and data [12,27,28,29]. Furthermore, trust and credibility are key aspects in overcoming these gaps and ensuring the successful implementation of IPC programs [1,2].

Resource limitations can potentially have a negative impact on the implementation of guidelines. In some cases, there are gaps in organizational infrastructure, poor laboratory capacity, infrastructure and data management challenges, use of non-prescription antimicrobial drugs, lack of public awareness, and insufficient general access to IPC. Lack of comprehensive national AMR programs can further compound these challenges [15,27,28]. The influence of trust and perceived credibility in the guidance and leadership can significantly affect how these challenges are navigated and how effectively guidelines are implemented [16,17].

#### 1.1.3. The Managers of the IPC Units

The managers of IPC units in hospitals during an epidemic require a range of skills and qualities to effectively implement and lead IPC programs. These include professional knowledge, strong management skills, autonomy, communication and training, leadership, power of persuasion, and the ability to deal with challenges and objections [12,15,28,29,30,31,32]. Moreover, establishing a strong trust relationship with their team members is crucial in enabling them to effectively lead and manage the IPC units, especially during an epidemic [1,2].

They also need to have strong communication skills, decision-making abilities, flexibility and adaptability, leadership, and empathy to make quick and effective decisions, lead teams, adapt to changing situations, and communicate effectively with team members, patients, and other stakeholders. Trust in their leadership and the credibility of their decisions plays a significant role in the team’s commitment and adherence to IPC practices [1,2].

The scope of their responsibilities is broad and includes controlling work processes, supervising services, interfacing with all levels and sectors, collaborating with parallel supervisory units, and assimilating and leading changes in the organization. Despite limited resources and the absence of structural and organizational infrastructure, IPC unit managers need to build a realistic implementation plan, make evidence-based guidelines available, provide regular updates and educational sessions to their team, and prioritize the program’s goals over competitive goals in the organization [12,15,28,29,30,31,32]. Trust in their leadership and credibility is crucial in driving these efforts towards success [1,2].

#### 1.1.4. The Sense of Self-Efficacy and Leadership in IPC

##### The Sense of Self-Efficacy

Self-efficacy, defined as an individual’s belief in their capacity to execute behaviors necessary to produce specific performance attainments [33], significantly affects motivation, behavior, and performance, particularly in challenging contexts [34,35]. In leadership roles within IPC practices, self-efficacy is vital because leaders who possess greater self-efficacy are more likely to influence and motivate their teams towards common goals [1,36]. Such leaders tend to exhibit proactive behaviors, are more inclined to take calculated risks, and could potentially drive innovation and organizational improvement [16,37]. Notably, self-efficacy is not immutable and can be developed through positive feedback, mastery experiences, and overcoming progressively challenging tasks [1,35].

##### Leadership

Leadership, characterized by the capacity to direct and influence others towards achieving a collective purpose [38,39,40], is significantly predicated on dynamic interactions and reciprocal influence [41]. In the domain of public health and IPC, trust in leadership becomes a pivotal element for ensuring adherence and implementation [16,42]. Effective leadership in IPC is about overcoming resistance and hesitance among healthcare staff by fostering efficient communication, trust-building, and acknowledging the emotional labor associated with compliance [2]. The willingness of healthcare workers to engage during crisis situations correlates directly with their trust in leadership [1]. Therefore, leaders in IPC must prioritize trust-building, communication enhancement, and the management of emotional responses to foster an efficacious IPC environment [11]. These facets of leadership become particularly salient during crises, such as epidemics or pandemics [43,44]. Leadership behaviors are fundamental to the successful adoption and sustainability of IPC measures within healthcare organizations [45,46,47,48]. The actions of leaders are more influential than their attitudes in the implementation of innovative IPC practices [49]. Especially during times of organizational change, behaviors that facilitate change acceptance and innovation are crucial [50,51,52,53,54].

##### Leadership’s Role in Infection Prevention

To prevent infections and mitigate the risk of antimicrobial resistance, leadership at the high level, middle level, and frontline is imperative. The success of IPC measures largely depends on the quality of leadership, the influence exerted, and the empowerment of others [55]. High-level leaders include those in hospital administration and heads of IPC units, who need to demonstrate high self-efficacy, address barriers effectively, and champion a culture of clinical excellence. These leaders should be adept, assertive, professional, and capable of advocating for resources and empowering frontline workers [4,15,23,56,57,58,59,60]. The gap in high-level leadership has widened due to insufficient awareness and support for IPC initiatives [15]. Middle-level leaders are instrumental in promoting a culture of safety and continuous quality improvement, encouraging learning from errors, and fostering open communication within a non-punitive environment. They are key in allocating resources and facilitating staff education [56,57,61]. Frontline leadership is essential for the day-to-day prevention of infections. Effective IPC programs often involve frontline staff as IPC champions, supported by IPC teams and senior leaders, which encourages a culture of collaborative leadership and psychological safety and minimizes implementation challenges [22,31,61,62,63,64,65]. Empowering frontline staff to uphold IPC standards has led to the emergence of informal leadership that contributes to solving local issues [62,66,67].

In summary, leadership at all levels is critical to the promotion and success of IPC initiatives. Cultivating psychological safety, effective communication, consistent feedback, and support from senior leadership can lead to improved IPC outcomes. Thus, further research is essential in this vital area of IPC [56,57,58,59,62].

##### Effective Leadership during the COVID-19 Pandemic

The COVID-19 pandemic has underscored the essential role of effective leadership and reinforced the importance of self-efficacy within IPC in healthcare settings. Leaders, especially at the helm of IPC units, have had to demonstrate flexibility to rapidly changing situations, ensure clear communication, manage resources judiciously, collaborate across disciplines, and proactively anticipate and address challenges [67]. Incidents of increased healthcare-associated infections during the COVID-19 pandemic have highlighted the necessity for improved IPC measures [21,22]. Additionally, the challenges of bacterial and fungal co-infections in COVID-19 patients have stressed the ongoing need for IPC leaders to refine their strategies [24]. Insights from national surveys, like those conducted in Israel, have pointed out the critical nature of adaptable IPC practices in response to evolving pathogens [23]. While IPC protocols are fundamental, the core of effective IPC is grounded in leadership approaches. Active leadership, creative communication strategies, and continuous feedback are recognized as key management practices for infection prevention [32]. It is also essential to equip frontline staff with the agility to adapt their practices in line with updated clinical guidelines. Moreover, building a culture of safety, encouraging teamwork, maintaining transparent communication, and having backup plans are integral for managing potential outbreaks or contamination issues [32,68]. Guidance from leading global organizations, such as the CDC and WHO, provides valuable direction [22,32,69], underscoring the complex challenges of IPC during a pandemic and the need for leaders to stay informed and ready to act. The responsibility of pandemic response extends beyond healthcare professionals to leaders in various sectors. For instance, during the 2009 H1N1 pandemic, effective leadership and heightened self-efficacy were linked to healthcare staff prioritizing patient care [48]. Research suggests that the readiness of emergency medical services to respond to pandemics, as well as routine situations, is significantly influenced by leadership quality and organizational structure [45,46,54]. Hence, the emphasis on management and leadership in the scholarly literature [70,71,72,73] highlights the importance of investigating the impact of COVID-19 on IPC unit managers within Israeli hospitals.

## 2. Materials and Methods

### 2.1. Participants

Utilizing a qualitative methodology, this study delved into the impacts of the COVID-19 pandemic on managerial self-efficacy, autonomy, and leadership capacities of IPC unit managers in Israel’s public hospitals. The qualitative approach was chosen for its ability to provide a holistic perspective into the personal experiences and viewpoints amid the unique challenges of the pandemic [74].

A total of 10 managers from nine distinct hospitals participated, representing 30% of the acute public hospitals in Israel. These roles included an equal distribution between 5 IPC unit physician managers and 5 IPC unit nursing managers.


**Demographics and Professional Background:**
Age: Participants had an average age of 54.6 years, ranging from 38 to 69 years.Professional Tenure: On average, they had worked in the medical profession for 29.4 years, with a span from 11 to 49 years.Tenure in IPC Unit: They had been serving in the IPC unit for an average of 14.6 years, with a range from 3 to 23 years.Managerial Experience in IPC Unit: Their managerial tenure within IPC units averaged 11.5 years, with experiences ranging from 2 to 23 years.



**Hospital Affiliation:**
Five participants managed IPC units in hospitals with over 800 beds, with three of these also overseeing units in hospitals with 200–400 beds.The remaining five managed IPC units in medium-sized hospitals with 400–800 beds.



**Interview Process:**
Interviews lasted on average 1 h and 42 min, with a range from 1 h and 20 min to 2 h.


These managers were chosen for their extensive professional experience and broad management history, providing a comprehensive perspective across various hospital sizes and dynamics.

### 2.2. Selection Criteria

Criteria for Inclusion: Active IPC unit managers within public hospitals were selected for this study.

Exclusion Criteria: No exclusion criteria were specified to ensure a comprehensive exploration of the field.

### 2.3. Sampling Method

Purposive sampling was employed to select a representative cohort of IPC unit physician and nursing managers, aimed at capturing a broad spectrum of insights into the managerial implications during the pandemic.

Table 1 conveys critical demographic and professional characteristics of IPC unit managers involved in the study, outlining the scope of their roles within different hospital capacities, as well as their gender, age, educational level, job commitment, overall professional experience, tenure in IPC units, and duration of managerial positions. This demographic backdrop serves as an instrumental reference for examining the effects of the COVID-19 pandemic on their managerial experiences, allowing for a nuanced analysis of how their backgrounds may have influenced their coping strategies and leadership effectiveness during the crisis.

Table 2 highlights the specialized training programs undertaken by participants, illuminating the diverse skill sets they possess. The prevalence of training in basic infection prevention and advanced management underscores the dual technical and leadership competencies deemed necessary for IPC management. These competencies were likely put to the test and further developed through the practical challenges and adaptive demands encountered during the pandemic.

Table 3 outlines the distribution of time across various managerial activities by the IPC unit managers over the past year. The higher score assigned to infection monitoring suggests a significant pivot in managerial focus towards direct pandemic response. Meanwhile, the consistent scores for management, training, and investigations reflect a sustained commitment to core IPC functions. This table may indicate the reallocation of managerial efforts in the face of shifting priorities caused by the pandemic, offering insights into the adaptive strategies employed by IPC unit managers to maintain operational efficacy during the crisis.

### 2.4. Instruments and Ethical Considerations

The interviews were designed to capture a detailed picture of the participants’ professional experiences throughout the pandemic. Open-ended questions were used to gain insights into the changes in their managerial capabilities, their sense of autonomy, and their leadership skills.

Post-interview, the data amassed were faithfully transcribed. Employing a thematic analysis methodology, these transcriptions underwent coding. This systematic approach encompassed several phases: thorough revisits of the interview content, inception of primary codes, identification of patterns and themes within the codes, a reflective overview of the unearthed themes, their subsequent crystallization, and the final compilation of the research report. Throughout this investigative journey, the imperatives of confidentiality, privacy, and anonymity were rigorously honored. All the participants were issued an informed consent form; thus, the unwavering commitment to ethical standards underscored the reliability and authenticity of the study’s findings and conclusions.

## 3. Results

### 3.1. Qualitative Findings

This chapter presents the findings of the research study conducted to explore the influence of the COVID-19 pandemic on managerial self-efficacy, autonomy, and leadership skills among managers in IPC units in public hospitals. The qualitative analysis revealed four key themes that illuminate various facets of the research goal. The themes identified via thematic analysis are:The impact of the COVID-19 pandemic on managerial self-efficacy, autonomy, and leadership skills;Management’s perception of the IPC units;Skills required for program implementation within the organization;Job satisfaction and personal well-being.

Each theme is detailed in the subsequent sections, bolstered by anonymized quotes from participants, offering profound insight into their experiences, viewpoints, and the transformations incited by the pandemic.

#### 3.1.1. The Impact of the COVID-19 Pandemic on Managerial Self-Efficacy, Autonomy, and Leadership Skills

Participants reported significant challenges during the initial months of the COVID-19 pandemic, highlighting feelings of isolation and uncertainty in their managerial roles due to a lack of information and conflicting guidelines. For instance, one manager shared, “The initial months of the COVID-19 outbreak were exceedingly challenging. I often felt isolated and confronted with uncertainty due to insufficient knowledge, conflicting directives, and hurdles in managerial capability and decision-making” (S2, D, 2 April 2023). This statement illustrates the initial obstacles faced by managers, such as feelings of isolation, uncertainty, and difficulty in making decisions due to limited information and inconsistent guidelines.

Another participant conveyed, “During the pandemic, hospitals felt like secluded islands. We had to innovate from the ground up. The confrontation with uncertainty led us to realize our capacity to adapt and tackle challenges on a larger scale. It was an amalgam of emotions—feeling forsaken yet simultaneously growing stronger” (S3, N, 4 April 2023). This reflection captures the sense of isolation within hospitals during the pandemic and the necessity to establish new protocols, highlighting a mixture of feelings including abandonment and empowerment through adaptation. These experiences align with the intensive training and the high proportion of time dedicated to infection monitoring and management reflected in Table 2 and Table 3, suggesting that the advanced management skills and the commitment to core IPC functions likely fostered a culture of resilience and adaptation among the managers.

#### 3.1.2. Management’s Perception of the IPC Units

Reflecting on the evolving role of IPC units within hospitals, participants noted a greater degree of support and understanding from upper management, albeit with an acknowledgment that there remains room for improvement. A participant stated, “Support from the management has increased, especially in recognizing the significance and role of the unit in managing the field, though there is still potential for enhancement” (S3, D, 3 April 2023). This implies that there has been an uptick in support from management, particularly in valuing the IPC unit’s role and its importance in field management, yet further improvement is desirable.

A participant noted, “There’s been a shift in management’s priorities. IPC, once considered a lower-tier concern, is now somewhat eclipsed by the hospital management’s focus on the human experience. I believe IPC is integral to all facets, as the human experience is directly tied to factors like patient density, cleanliness, and safe treatment, all within the realm of IPC” (S4, N, 9 April 2023). During the COVID-19 crisis, IPC garnered significant attention; however, as the acute phase of the pandemic subsided, other concerns emerged, shifting the focus despite the clear connection between IPC practices and a positive patient experience.

These perceptions are informed by the participant demographics in Table 1, where a substantial number of IPC unit managers hold advanced degrees and extensive experience in their field, underscoring the potential for these seasoned professionals to recognize and articulate shifts in managerial perspectives and priorities.

#### 3.1.3. Skills Required for Program Implementation within the Organization

The pandemic underscored the need for decisive leadership and the swift creation of new standards. Managers highlighted the significance of their decision-making skills, which became especially apparent during the pandemic: “I had to create standards without prior preparation to remain relevant in the decision-making process” (S3, D, 3 April 2023).

Furthermore, through hands-on experience, they acquired skills essential for navigating the crisis, including managing uncertainty and investigating outbreaks: “I honed skills in managing uncertainty, time management, and outbreak investigation through hands-on experience” (S2, N, 31 March 2023). This emphasis on direct experience aligns with the continuous professional development and specialized training programs engaged in by participants, as documented in the study’s tables.

#### 3.1.4. Job Satisfaction and Personal Well-Being

The pandemic’s impact on job satisfaction and personal well-being was profound. Managers articulated a change in their work experiences, characterized by a decrease in job satisfaction, yet an increase in commitment and motivation. The intense demands of their roles led some to reconsider their future in IPC management due to the emotional toll: “I was adversely affected, less fun at work, the change in perception has come to me, and I can no longer leave issues to be dealt with or to chance. Great commitment, a motivating source, before I could ignore things, today I am confronted with things I don’t like to deal with, and it is abrasive” (S3, D, 3 April 2023).

Another manager reflected on the mental and emotional exhaustion from the relentless pace of work: “The relentless investigations broke me down. It’s imperative to acknowledge that units need reinforcement; teams were overlooked, work seemed never-ending. The thought of IPC as managers, all the backstage efforts, our constant reinvention questions whether to continue in this work and role” (S5, N, 10 April 2023). This reflects the overwhelming nature of ongoing investigations, the need for stronger support structures, the unseen labor of managing, and the self-questioning about the sustainability of their role in the face of persistent challenges.

The narratives of these managers underline the dual-edge nature of the pandemic’s impact: while it brought about professional growth and strengthened resilience, it also led to increased stress and emotional burden. These findings are a testament to the complex psychological landscape that IPC unit managers have navigated during the pandemic, balancing the need for professional diligence with the care for personal well-being.

### 3.2. Probable Correlations in IPC Leadership Proficiency

Drawing on the qualitative data from Section 3.1.1, Section 3.1.2, Section 3.1.3 and Section 3.1.4, and reinforced by demographic and professional statistics, it is likely that the adeptness of IPC managers during the COVID-19 challenge stems from their substantial career history. The average professional tenure of 29.4 years, combined with an average of 14.6 years dedicated to IPC units and 11.5 years in managerial positions, suggests a strong foundation for their effective crisis response.

The demographic details in Table 1 highlight a diverse and seasoned group of leaders, with representation from various hospital sizes and extensive academic qualifications. This varied background likely contributes to their strategic and adaptable leadership, as indicated by their time allocation changes during the pandemic detailed in Table 3.

Table 2’s documentation of specialized training in key areas such as infection prevention and management underpins the managers’ dual competencies. This extensive training, paired with their lengthy tenures, suggests a high probability that these leaders have developed the necessary skill set for effective crisis management, honed through previous experiences in high-stakes health crisis scenarios.

### 3.3. Implications of the Pandemic on IPC Unit Management: Hypothesis Analysis

Transitioning from the qualitative findings presented in Section 3.1.1, Section 3.1.2, Section 3.1.3 and Section 3.1.4 to a more analytical discourse, it is crucial to revisit the original hypotheses posited at the outset of the research. These hypotheses were formulated to explore the complex impact of the COVID-19 pandemic on the roles and perceptions of IPC unit managers. The analysis of the qualitative data in relation to these hypotheses paints a multi-faceted picture that sets the stage for subsequent discussions and recommendations.

**Hypothesis** **1.**
*Managerial Self-Efficacy, Autonomy, and Leadership Skills.*


The hypothesis posited that the COVID-19 pandemic would significantly enhance managerial self-efficacy, autonomy, and leadership skills among managers in IPC units. The qualitative data underscored an amplification in managerial capabilities, autonomy, and leadership competencies as managers contended with the pandemic’s challenges. These accounts substantiate the hypothesis, indicating that initial adversities involving uncertainty and isolation were mitigated through the development of innovative processes and the embracement of adaptive strategies. The observed increase in self-efficacy and the fortification of leadership abilities are congruent with the expected positive outcomes.

**Hypothesis** **2.**
*Management’s Perception of IPC Units.*


It was hypothesized that hospital management’s perception of IPC units would become more positive, potentially resulting in augmented support and resources. Narratives from participants denote a perceptual shift within hospital administration, acknowledging the pivotal importance of IPC units. This change aligns with the second hypothesis, suggesting that the pandemic has likely engendered a more favorable viewpoint towards IPC units, culminating in enhanced support and allocation of resources.

**Hypothesis** **3.**
*Skills Required for Program Implementation.*


The hypothesis asserted that, post-pandemic, managers would recognize a distinct set of critical skills necessary for program implementation, emphasizing swift decision-making, adaptability, and crisis management. The imperative for resolute leadership and prompt adaptability during the pandemic has prompted managers to identify and hone a crucial skill set. These skills, highlighted by the respondents, corroborate the third hypothesis, indicating a reevaluation and advancement of the competencies deemed vital for effective program administration within their organizations.

**Hypothesis** **4.**
*Job Satisfaction and Personal Well-being.*


The hypothesis projected a dual impact on IPC unit managers’ job satisfaction and personal well-being due to the pandemic, with an expected dip in job satisfaction but a rise in a sense of achievement and purpose. The intricate emotional landscape extracted from the findings offers a nuanced perspective that resonates with the fourth hypothesis. Managers reported heightened stress and a dip in job satisfaction; nevertheless, they also conveyed an intensified sense of dedication and purpose, suggesting a partial affirmation of the anticipated dual effect on their well-being and satisfaction.

In concluding this chapter, we have delineated the congruence of the data with the initial hypotheses, thereby laying the groundwork for the ensuing discussion. The evidence accrued indicates a period of professional maturation for managers in IPC units, a redirection of organizational focus towards IPC operations, and the advent of novel managerial proficiencies. Concurrently, it underscores the personal costs associated with such brisk professional evolution, particularly regarding job satisfaction and mental well-being. These insights not only enhance our comprehension of the pandemic’s impact on IPC units but also inform future policy formulation and support structures for healthcare managers. The following chapter will delve into a deeper analysis of these implications and craft recommendations for continued research and practice.

## 4. Discussion

In this discussion, we critically examine the pandemic’s impact on IPC unit management through four primary hypotheses, scrutinizing the development of enhanced leadership capabilities, the transformed perceptions by management, the emergence of crucial skill sets for IPC program implementation, and the complex dynamics between job satisfaction and the well-being of IPC managers, along with additional findings that emerged from our study.

**Hypothesis** **1.**
*Managerial Self-Efficacy, Autonomy, and Leadership Skills.*


Our data support the proposition that the pandemic has enhanced managerial self-efficacy, autonomy, and leadership skills within IPC units. Crisis leadership demands self-assurance, decisiveness, and the independent capacity to lead in unprecedented situations. IPC managers have seen these skills sharpened under the pressures of the pandemic, accelerating their development. Their ability to quickly adapt, make informed decisions with limited information, and steer their teams through ambiguity has been critical [32]. The necessity for these leaders to navigate conflicting guidelines and rapidly changing scenarios has further highlighted the importance of self-efficacy and strong leadership [1,2,32,67]. An increased dependence on managerial autonomy signifies not just a practical response but also a measure of the trust and responsibility vested in these leaders by their institutions.

**Hypothesis** **2.**
*Management’s Perception of IPC Units.*


The study confirms that hospital management’s view of IPC units has grown more favorable, signaling a broader reassessment of the role of IPC within healthcare systems. Where IPC was once a routine consideration, the pandemic has thrust it into prominence [67]. This shift has prompted advocacy for more resources and greater acknowledgment of the critical role these units serve in hospital operations [12,27,28,29]. Such changes likely mirror an organizational paradigm shift, elevating IPC priorities in hospital management and strategic planning [67].

**Hypothesis** **3.**
*Skills Required for Program Implementation.*


In line with our third hypothesis, IPC managers have identified a new set of crucial skills necessary for effective program implementation. The demands of the pandemic have forged abilities in rapid decision-making, adaptability, and crisis management [67]. IPC managers often faced the need for immediate decisions made on partial information, with significant consequences [23]. This reality points to an IPC field where agility and foresight in crisis response are increasingly indispensable. The pandemic’s conditions required fast-track development in these areas. This is especially evident in the execution of comprehensive IPC programs, which revealed deficiencies in routine operations and underscored the need for solid infection control foundations [12,27,28,29,67].

**Hypothesis** **4.**
*Job Satisfaction and Personal Well-being.*


Our findings support the fourth hypothesis, indicating a nuanced impact of the pandemic on job satisfaction and personal well-being among IPC unit managers. While there’s a noticeable dip in job satisfaction due to increased stress, burnout, and the emotional toll of the pandemic [45,46,54], this is countered by a strengthened sense of achievement and purpose. This dichotomy aligns with the literature that points to the psychological rewards of overcoming challenges and contributing to public health [69]. The mixed outcomes highlight the need for comprehensive support for IPC managers, promoting resilience and well-being during and after such crises.

### Additional Discussion Points

Integrated Management Strategies in IPC Units: The pandemic has underscored the importance of strategic management in IPC units. The study reinforces the need for ongoing HAI surveillance and hospital diligence in infection rate monitoring. Incentive models proposed by health authorities could boost the maintenance of high-quality IPC practices, in agreement with strategies recommended in the literature to reduce HAIs [19,23]. Balancing the immediate demands of COVID-19 while sustaining robust IPC protocols is crucial to curtail hospital-acquired infections, particularly during infection surges [22,26].

Leadership Resilience and IPC Focus during the Pandemic: IPC managers had to demonstrate resilience and robust leadership during the pandemic. This study echoes research underscoring the criticality of leadership in times of crisis [32]. Managers adept in leadership and communication were especially capable of meeting the pandemic’s challenges [1,2]. The importance of leaders’ abilities to quickly adapt and make informed decisions, despite scant information, is a crucial skill set in crisis situations [32].

Efficacy of Comprehensive IPC Programs: Highlighting the gaps in IPC routine practices, the study emphasizes the necessity for comprehensive IPC programs, as developed by NPIPC and other global experts [21]. Effective infection control relies heavily on a solid organizational framework and leadership [12,27,28,29].

Trust and Credibility in IPC Execution: The effectiveness of IPC measures in managing resource constraints, addressing antimicrobial misuse, and enacting change depends greatly on trust and credibility within the IPC framework, as suggested by the literature [1,2].

The Central Role of Leadership in IPC: Finally, the pandemic has highlighted the crucial role of leadership within IPC units. Strengthening leadership approaches, supporting IPC programs, and recognizing the commitment of healthcare leaders are more vital than ever [56,57,59,62]. As the post-pandemic healthcare landscape evolves, the focus on advancing leadership capabilities to face future challenges effectively remains paramount.

## 5. Conclusions

The study emphasizes the vital role of strong leadership and effective IPC strategies in managing the COVID-19 crisis in healthcare. It points out that leadership quality and IPC are key in adapting to changes and ensuring quality care. The mental health of IPC staff is highlighted as crucial for maintaining efficiency.

The research also sheds light on the pandemic’s impact on healthcare workers’ satisfaction and well-being, noting increased commitment amidst fatigue and stress. It underscores the dual nature of the crisis as a barrier and a catalyst for professional development and the elevation of IPC roles in hospitals.

In summary, the pandemic has served as a testing ground for leadership and system agility, with the study advocating for future research to develop crisis-specific leadership models and enhance healthcare resilience.

## 6. Forward-Looking Statements

As healthcare systems reflect on the lessons learned from the COVID-19 experience, the call for visionary leadership becomes clear. The future of healthcare crisis management will likely prioritize the development of agile, adaptive leaders who are prepared for a range of complex scenarios. Institutions may also look towards fostering environments that prioritize mental health and employee well-being, recognizing their essential role in maintaining a functional and effective workforce during times of stress.

## 7. Study Limitations

Small Sample Size: The study involved only 10 managers from 30% of Israel’s acute public hospitals, which may not reflect the experiences of all hospital IPC unit managers.

Qualitative Focus: Being a qualitative study, the findings are not statistically generalizable but provide in-depth insights into the participants’ experiences.

Lack of Diversity: Limited gender diversity with 8 females and 2 males may influence the perspectives shared.

Potential Bias: As no exclusion criteria were defined, there may be biases in participant responses based on self-selection.

Pandemic Context: The unique circumstances of the COVID-19 pandemic mean that the findings may not be applicable to non-crisis situations.

## 8. Future Implications

The COVID-19 pandemic has fundamentally changed the role of IPC units, highlighting the need for strong leadership and swift decision-making. Looking ahead:

Strengthened IPC Role: IPC units will become central to both crisis response and everyday healthcare operations, requiring ongoing investment in resources and training.

Managerial Development: There will be an emphasis on cultivating leadership skills that cater to crisis management, adaptability, and rapid problem-solving.

Technological Integration: IPC will increasingly leverage technology, such as data analytics for infection trends and telehealth for patient care, necessitating new skills for IPC professionals.

Policy and Research Focus: Post-pandemic, expect a surge in IPC-focused research and policy development aimed at building resilient healthcare systems.

Global Health Security: IPC units will likely influence global strategies for health security, emphasizing collaboration and preparedness for future health threats.

In essence, the IPC landscape is set for evolution, marked by advanced preparedness, innovation, and a shift towards comprehensive health management.

## Figures and Tables

**Table 1 healthcare-11-02966-t001:** Description of Study Participants.

Feature	Value
Role	5, Doctor—Head of IPC Unit
	5, Nurse—Head of IPC Unit
Hospital Affiliation (>800 beds)	3 doctors, 2 nurses
Hospital Affiliation (400–800 beds)	2 doctors, 3 nurses
Gender	8 females, 2 males
Average age	54.6 (Range: 38–69)
Education	8 with Master’s/MD, 1 with Professorship
Job Percentage	100% for all 10 participants (5 nurses + 5 doctors)
Professional seniority	Average: 29.4 years (Range: 11–49)
Years of experience in IPC unit	Average: 14.6 years (Range: 3–23)
Managerial seniority in IPC	Average: 11.5 years (Range: 2–23)

**Table 2 healthcare-11-02966-t002:** Specific Training Undertaken by Participants.

Training Topic	Number of Individuals
Basic Infection Prevention Course/Specialization	9
Advanced Management Skills	7
Infection Monitoring and Outbreak Investigation	6
Development and Promotion of Work Programs	4
Implementation of Organizational Changes	2

**Table 3 healthcare-11-02966-t003:** Participants’ Time Allocation to Various Activities Over the Past Year.

Activity	Average Score
Management, team promotion, and work plans	3.7
Training and guidance	3.5
Audits/Observations/Consultations	3.6
Infection monitoring	4.3
Investigation of incidents and outbreaks	3.6
Research	2.6

## Data Availability

Data are contained within the article.

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
