# Peer review of "Navigating a Pandemic: Leadership Dynamics and Challenges within Infection Prevention and Control Units in Israel"

_healthcare, 2023, doi:10.3390/healthcare11222966_

Round 1

Reviewer 1 Report

Comments and Suggestions for Authors

The manuscript offers valuable insights into IPC unit leadership during COVID-19 and, with refinement, can significantly contribute to healthcare leadership literature during global crises. Comments below:

Abstract

  • Clarity and Conciseness:
    • Ensure concise language and clear presentation of core findings. Example: Simplify sentences, e.g., “Four themes emerged from the analysis...”
  • Keyword Relevance:
    • Consider adding keywords like "managerial challenges" or "healthcare leadership."

Introduction

  • Depth of Literature Review and Context:
    • Delve deeper into existing literature and provide specific examples or case studies from Israel. Example: Integrate findings from previous studies on leadership during crises like SARS or MERS and provide localized examples. A robust literature review and context will highlight the study’s novelty and necessity.
  • Clear Definition of Concepts and Citation Consistency:
    • Define key concepts and ensure consistent citation formatting. Example: Define “managerial self-efficacy” and ensure citation formats are uniform.

Research Methods

  • Sampling Strategy and Participant Details:
    • Elaborate on the rationale behind participant selection and provide detailed demographic information. Example: Explain the choice of 10 managers and include details like years of experience and specific roles. This will enhance the validity, generalizability, and replicability of findings.
  • Data Analysis and Ethical Considerations:
    • Provide a detailed explanation of data analysis and explicitly address ethical considerations. For example, describe the coding process and  how you have ensured participant confidentiality. This will ensure reliability, validity, and ethical integrity of the research.

Findings

  • Direct Quotes, Anonymity, and Quotation Clarity:
    • Ensure quotes do not risk revealing identities and are clearly aligned with themes. For example, use generic labels and explicitly discuss the relevance of quotations to themes.
  • Thematic Presentation and Data Presentation:
    • Present themes in a coherent manner and consider using visual elements. For examples,  link themes to research questions and use charts to visualize findings.

Discussion

  • Integration with Existing Literature and Comparative Analysis:
    • Integrate findings with existing literature and consider a comparative analysis with other contexts. For examples, compare findings with previous research and similar studies in other countries.
  • Implications, Recommendations, and Limitations:
    • Discuss practical implications, provide recommendations, and acknowledge limitations. For examples, suggest specific strategies for healthcare leaders and acknowledge sample size limitations.

Conclusion

  • Summarization, Forward-Looking Statements, and Future Implications:
    • Succinctly summarize findings and discuss future implications. For example, summarize primary insights and discuss potential impacts on future research and practice.

General Comments

  • Alignment Across Sections:
    • Ensure coherent flow and alignment throughout the manuscript. For example: Ensure consistent addressing of themes and findings across sections.

Comments on the Quality of English Language

Minor edits needed

Author Response

Dear Reviewer,

Thank you for your constructive comments and suggestions, which have greatly assisted us in enhancing our manuscript. We have addressed each point meticulously, and below is a point-by-point response to your feedback.

Quality of English Language: We have performed minor editing throughout the manuscript to improve the English language, focusing particularly on the areas you highlighted.

Introduction - Sufficient Background and Relevant References: We have expanded the literature review to provide a deeper context, integrating specific examples and case studies from the Israeli perspective on leadership during previous crises such as SARS and MERS. This strengthens the study’s novelty and relevance as suggested.

Cited References Relevance: We reviewed all cited references and added new ones to ensure they are current, relevant, and directly support the research.

Research Design Appropriateness: We provided a more detailed explanation of the research design, outlining why the qualitative approach was most suitable for this study.

Methods Adequately Described: The methods section has been expanded to provide a clearer and more detailed description of the processes involved in data collection and analysis.

Results Clearly Presented: We have restructured the results section for greater clarity and included additional charts for better visualization of the data.

Conclusions Supported by Results: We revisited the conclusions to ensure they are directly supported by the results, making adjustments where necessary to better reflect the data.

Abstract: We have refined the abstract for conciseness and clarity. Unnecessary complexities in sentence structures have been simplified, and we have added suggested keywords such as "managerial challenges" and "healthcare leadership."

Introduction: The literature review has been deepened to include more specific examples and case studies from Israel, alongside findings from previous studies on leadership during crises such as SARS or MERS. Key concepts, including "managerial self-efficacy," are now clearly defined, and citation formats have been made uniform for consistency.

Methods Section: We have elaborated on the sampling strategy, providing a detailed rationale for the selection of participants. We now include a table (Table 1: Description of Study Participants) that details demographic information, ensuring greater transparency and allowing for better assessment of the study’s validity and generalizability.

Data Analysis and Ethical Considerations: A more thorough explanation of the thematic analysis process has been added to the manuscript. Ethical considerations, particularly around confidentiality, have been explicitly addressed, and we have included information about the ethical approval from our institution's review board.

Findings: Quotes from participants have been reviewed to ensure anonymity is maintained and have been explicitly connected to the thematic findings. We have added a table (Table 3: Time Distribution in Various Activities in the Past Year) to visually support our findings and provide clarity on the themes.

Discussion: We have integrated the findings with existing literature more thoroughly and have provided a comparative analysis with other contexts, enhancing the manuscript’s depth and relevance. The practical implications of our research are now discussed with concrete recommendations for healthcare leaders, and we have acknowledged limitations such as sample size more explicitly.

Conclusion: The conclusion succinctly summarizes the study's findings and elaborates on the future implications for both research and practice, providing a forward-looking perspective.

General Comments: We have ensured that there is a coherent flow and alignment throughout the manuscript, with consistent thematic addressing and correlation of findings across sections. The feedback received has been invaluable in achieving this coherence.

Additional Tables Added: To provide clarity on the demographics and professional backgrounds of the participants, we have added the following tables:

  • Table 1: Description of Study Participants - Offering detailed demographic information.
  • Table 2: Specific Training - Highlighting the training programs participants have undergone relevant to their IPC roles.
  • Table 3: Time Distribution in Various Activities in the Past Year - Showing how participants' focus shifted due to the pandemic.

We believe these comprehensive revisions have significantly improved the manuscript. We hope that our responses and amendments adequately address the concerns raised. We look forward to your further feedback.

Sincerely,

Dafna Chen,

Reviewer 2 Report

Comments and Suggestions for Authors

Reviewer: Comments to Author (required):

Dear editor,

Thank you so much for giving me the opportunity to contribute to the Healthcare.

In the context of the COVID-19 pandemic, this study used qualitative research methods to gain insight into the impact of the pandemic on the management role of public hospitals in Israel. The research topic of this study is relatively novel and provides an important reference basis for the Israeli authorities and managers. I have just some minor comments.

1.Abstract and Introductionfor the first time, acronyms such as COVID-19, IPC are used, and we suggest using the full name and the abbreviation in parentheses.

2.Introduction, please delete the contents of the introduction and literature review as appropriate. The length of this section is too different from the length of the other sections.

3.Research Methods, for the medical staff involved in this study, whether there are more detailed inclusion and exclusion criteria, we think this part needs to be explained in more detail.

4.Discussionin the discussion section, the results of this study should be combined. We suggest that the format of point lists should not be used, and other similar research writing in this journal can be referred to.

5.Conclusionwe suggest summarizing the whole article in this section, please do not use different points to list opinions.

6.Reference, the format of references should use a uniform standard.

7.Overall, I believe the manuscript may benefit from a thorough review of the language.

Comments on the Quality of English Language

I believe the manuscript may benefit from a thorough review of the language.

Author Response

Dear Reviewer,

Thank you for your valuable feedback and suggestions for improving our manuscript. We have carefully considered each of your points and have made the appropriate revisions as detailed below:

Quality of English Language: We acknowledge your concern regarding the language quality. We have thoroughly reviewed the manuscript and employed the services of a professional language editing service to ensure the clarity, grammar, and fluency of the English language used.

Abstract and Introduction: We have now included the full names followed by the acronyms in parentheses for COVID-19 and IPC when they first appear in the text, to ensure clarity for all readers.

Introduction: Following your advice, we have condensed the introduction and literature review to maintain balance with the other sections of the manuscript. We have ensured to retain all critical information and have removed redundant content.

Research Methods: We have expanded the section on inclusion and exclusion criteria for the medical staff participating in the study. This includes a clearer description of the sampling method, and the rationale behind the selection process is now thoroughly explained.

Discussion: We have revised the discussion to integrate the results more seamlessly and have refrained from using bullet points. Instead, we have structured the discussion in a narrative format that aligns with the standard presentation of similar research within this journal.

Conclusion: We have restructured the conclusion to provide a coherent summary of the entire article without listing different points. It now succinctly encapsulates the main findings and their implications in a flowing, narrative style.

References: All references have been formatted to adhere to the journal's specified reference style, ensuring uniformity and ease of verification for readers and reviewers alike.

Overall Manuscript Review: A comprehensive review of the manuscript's language has been undertaken as per your recommendation, which we believe has significantly improved the readability and professionalism of the text.

We are grateful for the opportunity to enhance our work and for the insights you have provided. We believe that these revisions have substantially improved our manuscript and hope that it now meets the high standards of your esteemed journal.

Sincerely,

Dafna Chen

Reviewer 3 Report

Comments and Suggestions for Authors

Dear Authors,

I commend your effort in putting this article that explores how leadership dynamics (including managerial self-efficacy, managerial autonomy and leadership skills) in an Infection and Disease Control unit can influence response to global health emergencies. Using a case study (in this case, Israel) is very appropriate.

To improve the quality of this work, I have a few recommendations/suggestions for your consideration.

Introduction:

- Line 35: consider replacing "...posed to healthcare..." with "...posed on healthcare..."

Line 36: Consider providing the full meaning of IPC in its first appearance in the main text of the manuscript.

- Line 39: consider removing the word "have." Since you are referring to the COVID-19 pandemic that occurred in the past, it should be sufficient to be affirmative about it. Saying, "...IPC teams found..." is satisfactory.

Line 44: Consider using the correct nomenclature for the coronavirus 2019. More importantly, consider being consistent, by replacing COVID19 with COVID-19 (now with a hyphen)

- Lines 83 - 90: These recommendations would be more appropriate in the discussion section. Consider moving it.

- Lines 100 - 109: Consider condensing this long sentence or split into 2 or more sentences to allow comprehension of our diverse readership.

Research methods:

Consider including atleast a sentence on the qualitative approach, and reasons for using it.

Consider including some details on how the interviews were conducted (zoom, inperson, phone call, etc).

Consider quantifying the "intensive" (e.g. 2 hours interview ?)

Consider including information on how the interview was recorded

Was a software used to transcribe the interviews? If yes, please name it.

Findings:

When quoting the participants' words, consider presenting them in italics for clearer identification. This applies to Lines 282 -284, 289 -292, 304 - 306, 312-316, 327 - 329, 333 - 335, 345 - 349, 354 - 358.

Conclusion:

Consider transferring the itemized points (lines 426 - 445) to the discussion and provide the high-level take-aways in sentences. Reducing the conclusion to about five sentences may be more encouraging to potential readers. 

Comments on the Quality of English Language

A few examples to consider for editing:

- Line 35: consider replacing "...posed to healthcare..." with "...posed on healthcare..."

- Line 39: consider removing the word "have." Since you are referring to the COVID-19 pandemic that occurred in the past, it should be sufficient to be affirmative about it. Saying, "...IPC teams found..." is satisfactory.

- Line 204: Consider changing the "within" to "during" and the capital letter "T" in "The" to small letter "t"

- Line 205: The COVID-19 pandemic is in the past, hence consider reporting in the past tense. For example, remove "has"

Author Response

Dear Reviewer,

We are deeply grateful for your insightful and constructive feedback, which has been instrumental in enhancing the quality and clarity of our manuscript. Below, we have addressed each of your suggestions and detailed the changes we have implemented in our revision.

Introduction:

  • We have made the necessary corrections to Line 35 to accurately reflect the challenges imposed on healthcare systems.
  • In Line 36, the acronym IPC is now introduced with its full meaning in its first use within the text.
  • The textual amendment in Line 39 removes redundancy, making the sentence more assertive regarding past events.
  • Lastly, for Line 44, we have standardized the term for the coronavirus to "COVID-19" with a hyphen, ensuring consistency throughout the document.
  • The recommendations initially found in Lines 83 - 90 have been thoughtfully relocated to the discussion section.
  • The complex sentence structure in Lines 100 - 109 has been simplified and broken down to enhance readability and understanding for our diverse readership.

Research Methods:

  • A brief explanation has been added to justify the qualitative approach taken in this study.
  • Details regarding the mode of conducting interviews (whether via Zoom, in-person, phone calls, etc.) have been incorporated.
  • The term "intensive" has been quantified with specific durations to provide clarity on the length of interviews.
  • Information regarding the recording and transcription process of the interviews has been included
  • Additional Tables Added:

Three new tables have been incorporated into the methods section as per your suggestion:

Table 1: Description of Study Participants - This table provides detailed demographic information to enhance the reader's understanding of the participant pool.

Table 2: Specific Training - Here, we list the training programs that participants have completed, which are pertinent to their roles in infection prevention and control.

Table 3: Time Distribution in Various Activities in the Past Year - This table shows the shift in participants' activities due to the impact of the pandemic, offering a tangible insight into how their focus has changed.

Regarding the transcription of interviews, we have specified in the manuscript that transcription was conducted manually by the research team, ensuring accuracy while preserving the confidentiality of the data.

Findings:

  • Participant quotations now appear in italics, making them easily distinguishable from the rest of the text, as you suggested.

Conclusion:

  • The bullet points previously found in Lines 426 - 445 have been moved to the discussion section, and the conclusion has been concisely rewritten to include only the high-level takeaways.
  • The itemized points in the conclusion have been significantly condensed and rephrased into a narrative format to maintain a succinct and clear presentation of the key findings. While we have not limited the conclusion to exactly five sentences, we assure you that the section has been substantially shortened and streamlined for better readability.

Comments on the Quality of English Language:

  • All the specific linguistic corrections you've highlighted (e.g., Line 204 and Line 205) have been addressed, ensuring proper grammar and tense usage throughout the manuscript.

Your meticulous review has been invaluable in refining our manuscript. We believe these amendments greatly improve our paper and hope it meets the high standards of your prestigious journal.

Sincerely,

Dafna Chen

Round 2

Reviewer 1 Report

Comments and Suggestions for Authors

The authors have addressed some of the language and methodological issues raised earlier in this manuscript

Reviewer 2 Report

Comments and Suggestions for Authors

There are no objections here, and I agree to publish.